# Mathematical model for the role of multiple pericentromeric repeats on heterochromatin assembly

**Puranjan Ghimire**[1], **Mo Motamedi**[2]*, **Richard Joh**[1,3]*

**1** Department of Physics, Virginia Commonwealth University, Richmond, Virginia, United States of America, **2** Massachusetts General Hospital Center for Cancer Research and Department of Medicine, Harvard Medical School, Charlestown, Boston, Massachusetts, United States of America, **3** Massey Cancer Center, Virginia Commonwealth University, Richmond Virginia, United States of America

* mo_motamedi@hms.harvard.edu (MM); johi@vcu.edu (RJ)

**Data Availability Statement:** The code will be made available on FigShare(https://figshare.com/articles/media/Untitled_Item/24599379).

**Funding:** RJ was supported by Institutional Research Grant IRG-18-159-43 from the American

## Abstract

Although the length and constituting sequences for pericentromeric repeats are highly variable across eukaryotes, the presence of multiple pericentromeric repeats is one of the conserved features of the eukaryotic chromosomes. Pericentromeric heterochromatin is often misregulated in human diseases, with the expansion of pericentromeric repeats in human solid cancers. In this article, we have developed a mathematical model of the RNAi-dependent methylation of H3K9 in the pericentromeric region of fission yeast. Our model, which takes copy number as an explicit parameter, predicts that the pericentromere is silenced only if there are many copies of repeats. It becomes bistable or desilenced if the copy number of repeats is reduced. This suggests that the copy number of pericentromeric repeats alone can determine the fate of heterochromatin silencing in fission yeast. Through sensitivity analysis, we identified parameters that favor bistability and desilencing. Stochastic simulation shows that faster cell division and noise favor the desilenced state. These results show the unexpected role of pericentromeric repeat copy number in gene silencing and provide a quantitative basis for how the copy number allows or protects repetitive and unique parts of the genome from heterochromatin silencing, respectively.

## Author summary

Pericentromeric repeats vary in length and sequences, but their presence is a conserved feature of eukaryotes. This suggests that the repetitive nature of pericentromeric sequences is a functional feature of centromeres, which is under selective pressure. Here we developed a quantitative model for gene silencing at the fission yeast pericentromeric repeats. Our model is one of the first models which incorporates the copy number of pericentromeric repeats and predicts that the number of repeats can solely govern the dynamics of pericentromeric gene silencing. Our results suggest that the repeat copy number is critical for gene silencing, and copy-number-dependent silencing is an effective strategy

Cancer Society. MM was supported by an ACS Research Scholar grant (18-056-01-RMC) and R01 (GM125782). Authors declare no competing financial interests, and the funders had no role in study design, data collection and analysis, decision to publish, or preparation of the manuscript.

**Competing interests:** The authors have declared that no competing interests exist.

to repress the repetitive part of the genome and protect the unique part of the genome from heterochromatin silencing.

## Introduction

The eukaryotic centromere is comprised of a core region that is flanked by multiple repetitive DNA sequences, called pericentromeric repeats. Pericentromeric repeats in eukaryotic cells are organized in a dense chromatin structure known as heterochromatin which is vital for the establishment and maintenance of chromosomal stability. For example, pericentromeric heterochromatin is essential for accurate chromatin segregation and genome stability [1,2]. Heterochromatin assembly is also required for gene silencing and repression of recombination among repeats [3]. The de-repression of pericentromeric repeats in human cancers (e.g., BRCA1-mutated breast cancer [4,5]), and tumor formation caused by the forced transcription of pericentromeric satellite RNAs in mice [6] suggest that the transcriptional silencing of pericentromeric repeats is also crucial for preventing tumorigenesis. Heterochromatin is often characterized by DNA methylation, repressive histone modifications, and hypo-acetylation of histones. One of the hallmarks of heterochromatin formation in eukaryotic cells is the methylation of lysine 9 (K9) of histone 3 (H3) [7–9]. This chromatin mark and the associated enzymes are conserved from a unicellular eukaryote, like the fission yeast, to a complex mammal, like a human [10].

In the fission yeast, there is one histone methyltransferase for H3K9, Clr4, which facilitates H3K9 mono- (me1), di- (me2), and tri- (me3) methylation. The nucleation of pericentromeric heterochromatin and its spreading by various chromatin-dependent mechanisms is extensively studied in fission yeast where the RNA interference (RNAi) pathway is predominantly responsible for pericentromeric heterochromatin assembly [11–14]. RNAi recruits Clr4 via the RNA-induced transcriptional silencing (RITS) complex physically interacting with Clr4 [15–18]. RITS is a small interfering RNA (siRNA)-containing effector complex, consisting of three proteins, the sole Argonaute homolog (Ago1), the GW domain protein, Tas3, and a chromodomain-containing protein, Chp1. RITS interacts with H3K9me via chromodomain protein Chp1 and nascent RNA via Ago1 [11–14]. In addition to the interaction with Clr4, RITS can also interact with RNA-dependent RNA polymerase complex (RDRC) [16,19], which together with Dicer (Dcr1) amplifies and processes nascent long non-coding RNAs (lncRNAs) into siRNAs creating a positive feedforward loop, where siRNAs increase H3K9me, and H3K9me stimulate the subsequent synthesis of siRNAs [12].

Although the silencing of pericentromeric repeats is a conserved feature of eukaryotes, there is a considerable variation among those repeats in terms of their lengths, constituting sequences, and organization [20]. However, the presence of multiple repeats, which flank the central core is a conserved feature [21]. This suggests that the repetitive nature of pericentromeres may be an evolutionary conserved feature with a functional consequence. Here, we hypothesize that the copy number of the repeats is critical for pericentromeric gene silencing. Experimental manipulation and quantification of copy number-dependent gene silencing are difficult because of the presence of identical repeats in the pericentromeric region. In this work, we developed a mathematical model for copy-number-dependent gene silencing in pericentromeric repeats, and our simple model supports that multiple copies are required for gene silencing. We have analyzed the robustness of model parameters through sensitivity analysis and stochastic simulation using the Gillespie algorithm. Our results suggest that the copy

number of pericentromeric repeats alone can significantly impact gene silencing, which may affect genomic stability and human diseases.

## Results

### Mathematical modeling of gene silencing at pericentromeric repeats

To investigate how the copy number of pericentromeric repeats affects gene silencing, we developed a mathematical model incorporating lncRNA, siRNA, and H3K9 methylation as key parameters involved in the establishment of heterochromatin. A schematic representation depicting a simplified molecular pathway used for mathematical modeling is shown in Fig 1, and the parameters are shown in S2 Table. Briefly, the model tracks the level of RNA (pericentromeric lncRNA), siRNA, and H3K9 methylation which provide the specificity needed for the recruitment of downstream complexes. These molecules also create H3K9me and siRNA feedforward loops, critical for the establishment of heterochromatin at pericentromeres (see Methods).

**RNA.** According to the nascent transcript model, lncRNAs provide a platform to facilitate siRNA-mediated H3K9me with RITS and Clr4 complexes [16–18]. Their transcription and recognition by RITS complex nucleate heterochromatin at centromeres. The lifetime of lncRNA was assumed to be 50min (S2 Table), based on the experimental data in yeast [22,23]. RNA can be degraded or turned into siRNAs in a methylation-dependent manner.

**siRNA and H3K9me amplification loops.** RITS contains Chp1, a chromodomain protein, capable of binding to H3K9me. This tethers the RITS-RDRC-Dcr1 siRNA biogenesis pathway (assembled on heterochromatin lncRNAs) to heterochromatin via siRNA base-paring interactions with the complementary nascent pericentromeric lncRNAs. Also, Clr4 physically

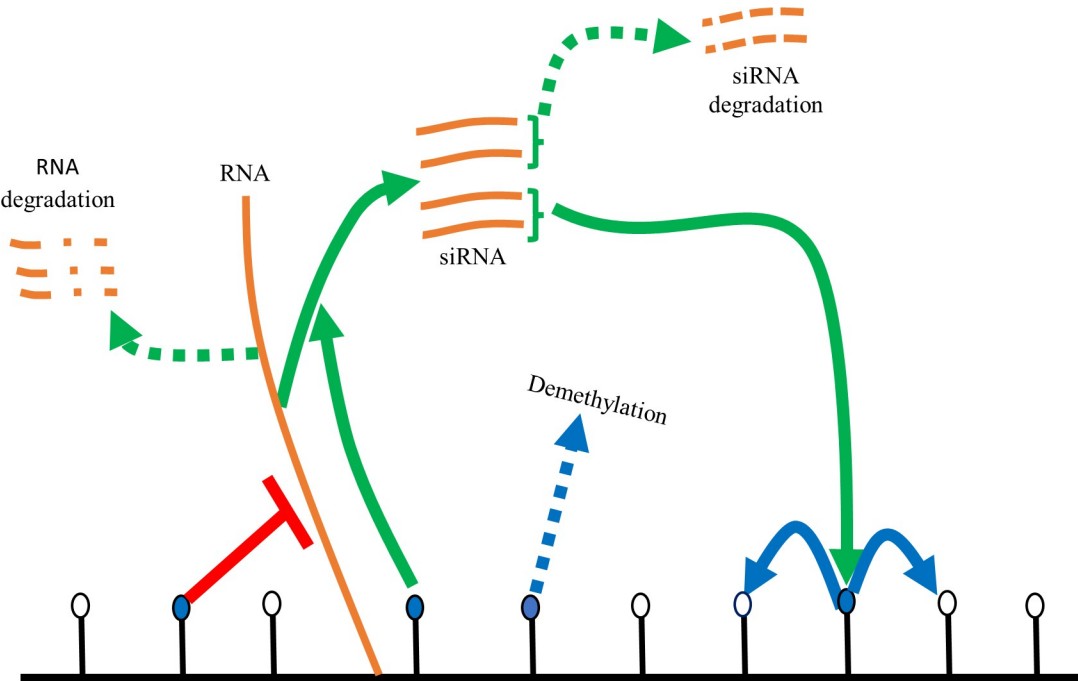

**Fig 1. Schematic diagram for mathematical modeling of siRNA-mediated gene silencing.** Blue and white circles represent methylated and unmethylated histones, respectively. siRNA can recognize nascent lncRNAs, which can lead to H3K9me, which can spread to nearby histones. H3K9me in turn represses the transcription, and RNA is either degraded or turned into siRNA.

interacts with RITS [15] and this interaction is critical for targeting Clr4 to heterochromatin. Once at heterochromatin, Clr4-mediated H3K9 methylation can spread in *cis*, and iterative cycles of siRNA generation, and RITS-Clr4-mediated H3K9 methylation amplify the H3K9me and siRNA signals at heterochromatin. H3K9me, in turn, represses the transcription [24], but even a small number of RNA can be efficiently turned into siRNA.

**Repeat copy number.**    There are over 10 copies of pericentromeric repeats in fission yeast. At least one repeat is present on each side of three S. pombe centromeres. The number of repeats varies across chromosomes as chromosome 3 has multiple copies on both sides of the centromere [25]. Each copy can have its own transcriptional activity independent of one another. However, Ago1-bound siRNA can target nascent transcripts across chromosomes and exogenous loci [26], and repeats show similar profiles in gene silencing and H3K9 methylation across chromosomes. The copy number is an explicit parameter in our mathematical modeling. Each copy of the pericentromeric repeat can be transcribed independently, and without any feedback, more copies promote more transcription linearly dependent on the copy number.

**Histone deacetylase activity.**    Although our model focuses on the H3K9 methylation, histone deacetylases (HDACs) are critical in the regulation of pericentromere silencing. Sir2 and Clr3 are HDACs in fission yeast, whose deletion leads to a partial loss of silencing [26–32]. The main substrate of Clr3 is H3K14ac, but its enzymatic activity is required for heterochromatin silencing via H3K9me [33,34]. To incorporate this process, we developed an alternative model (see Methods) to track three H3K9 states: 1) acetylated, 2) unmodified, and 3) methylated. We assume that the new histones are acetylated [35], and acetylated histones need to be unmodified prior to possible methylation [36].

## Bistability between silenced and desilenced states is possible at a small copy number

To understand the role of copy number on gene silencing, we first analyzed solutions of the ordinary differential equations (ODEs) (see Eqs (1–3)) while varying the copy numbers of pericentromeric repeats (CN). Our model assumes that repeats are transcribed and their copy number, an explicit parameter in our model, enhances transcription linearly for a given level of methylation. Fig 2A–2B depicts the dynamics of methylation from multiple initial conditions for different CNs, and S1 Fig shows the results of RNA, siRNA, and H3K9me. Specifically, there can be one (CN = 15) or two (CN = 5) stable steady states depending on the copy number. As described above, H3K9me modulates the lncRNA transcription via the RNAi machinery. If the copy number is low, the amount of siRNA is low, and the system remains desilenced (Fig 2A and S1 Fig). If the copy number is high, enough RNA and siRNA can be generated, which in turn silences the repeats through methylation (Fig 2B and S1 Fig). If H3K9me is low, RNA degradation proceeds via the conventional mechanism with few siRNAs, whereas if H3K9me is high, RNA is predominantly processed into siRNA.

To assess the effect of copy number change on gene silencing, we systematically varied the copy number and quantified the number of stable steady states. Fig 2C depicts the steady states of methylation for various CNs, where the blue and red curves represent the higher and lower steady states of H3K9me, respectively. Fig 2C shows that for copy numbers up to around 11, the system displays both silenced and desilenced steady states. However, for CN greater than 11, there is only one steady state, which is silenced. S2 Fig shows the dynamics from multiple initial conditions as CN increases, where the fraction of desilenced states decreases until the system becomes monostably silenced. The known copy number of lncRNA dg/dh in *S. pombe* wild-type (WT) cells is around 15, which is represented by the green star in Fig 2C, and the

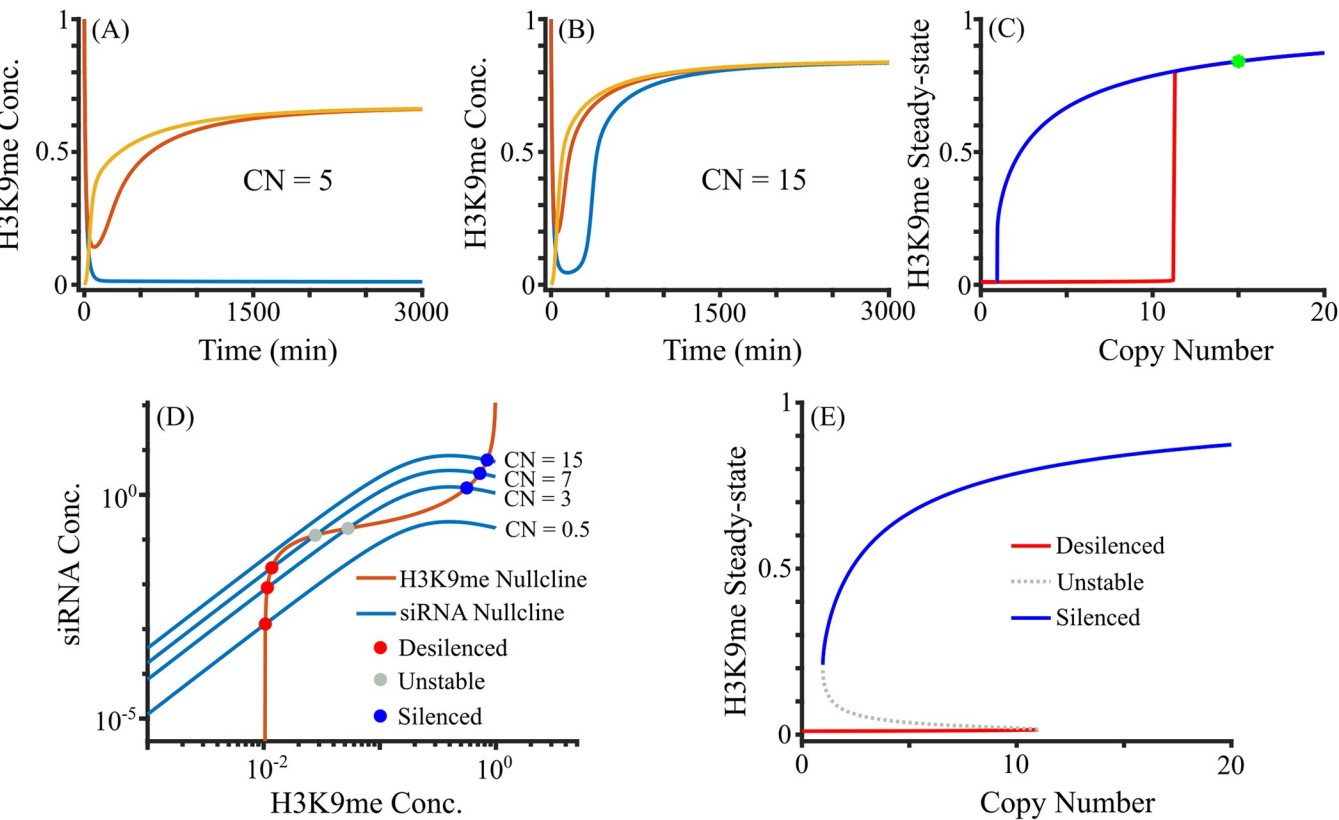

**Fig 2. Change in the steady-state behavior as a function of repeat copy number.** (A-B) Solutions of H3K9me for different $CN_s$ in the full model. Colors indicate the use of different initial conditions. (C) Steady-state H3K9me as a function of copy number. The system is bistable (silenced and desilenced) or monostable. Wild Type (WT) in fission yeast is shown at $CN = 15$ by a green star. (D-E) Quasi-steady state approximation. (D) Nullclines of siRNA and H3K9me using QSSA. Representative siRNA nullclines at indicated $CN_s$ are shown. (E) Bifurcation diagram showing the bistability with both silenced and desilenced states for smaller copy numbers and monostability at high copy numbers.

pericentromere of WT cells are silenced. Our results suggest that the heterochromatin-mediated silencing is nonlinearly dependent on the repeat copy number, which can change qualitatively from monostability to bistability. Overall, our model demonstrates that the repeat copy number is a critical dynamical parameter of RNAi-mediated silencing at the fission yeast centromeres.

## A small change in copy number leads to a big change in steady states of H3K9me

The turnover rate of siRNA is typically lower than RNA due to binding by Argonaute proteins [37,38], which results in slower changes in siRNA than RNA. Because RNA reaches steady-state faster than siRNA, to test how CN modulates the steady states of the system, we used a quasi-steady-state approximation (QSSA), in which we assumed that RNA concentration reaches equilibrium at any given siRNA and H3K9me level. The time evolution of siRNA and H3K9me in the full model is similar to that in the QSSA (S3 Fig). QSSA reduces three variables in the full model into two variables, which allowed us to plot nullclines in 2-dimensions and thus find the steady-states. Fig 2D shows the nullclines from Eqs (6–7), which represent curves where siRNA and H3K9me concentrations are not changing, respectively. The intersections of these two nullclines are the steady-states, which include both unstable and stable steady-states.

The number of steady states can change between 1 and 3, with 1 or 2 stable steady states. The siRNA nullcline is CN-dependent as the nullcline moves up with high CN, and such changes control the number of steady states of the system. The H3K9me nullcline is CN-independent but the nonlinear shape of this nullcline allows it to have one to three intersections. The intersection point with low methylation ($< 0.2$) is a desilenced steady state, and the intersection point with high methylation ($\sim 1$) is a silenced steady state. If there are three steady states, the middle steady state is unstable. For copy number 15, the siRNA nullcline intersects the H3K9me nullcline at the silenced state only, and this transition from bistability to monostability is due to the saddle-node bifurcation [39].

Fig 2E shows the bifurcation diagram, revealing the stable and unstable steady states as a function of the copy number. The system is bistable for lower copy numbers and becomes silenced via the saddle node around copy number 11. Between copy numbers 11 and 12, the system drastically changes its steady-state dynamics between a single silenced state and the coexistence of both silenced and desilenced states. A big change in steady states of H3K9me with a small change in copy number suggests that the copy number is an important factor for the gene silencing of cells, especially if cells were initially desilenced.

## Sensitivity analysis

To investigate the robustness of silenced and desilenced states in a cell, we systematically varied parameters and tracked how the behavior of the system changes. Most of the parameters in our model are related to the proteins participating in the RNAi pathway of pericentromeric gene silencing. To identify the key parameters in copy number-dependent gene silencing, we tracked steady-state changes when a parameter was varied within 25% of its reference value while keeping all other parameters fixed at their reference values. In all cases, such changes in the parameters can lead to 1) bistability between silencing and desilencing and 2) monostability of silencing or desilencing. Several factors in our model can repress methylation like demethylation and RNA degradation, while other factors like methylation by siRNA, siRNA biogenesis, and spreading of methylation enhance methylation. Fig 3A–3D shows if the solution is silenced or bistable at the indicated copy number and four representative parameters (see S4 Fig for other parameters). Enhancing anti-silencing processes can destabilize silencing and lead to switching from silencing to bistability. For example, a high RNA degradation ($\delta_1$) and demethylation rate ($\delta_3$) facilitate switching from silencing to bistability (Fig 3A–3B). On the other hand, enhancing H3K9me-associated processes can stabilize silencing. Fig 3C–3D shows the changes in the steady state behaviors where a high methylation rate ($\varepsilon$) and methylation spreading rate ($\phi$) favor silencing.

To test how each parameter affects system behavior, we also analyzed how the nullclines change from QSSA when one of the parameters is changed. Fig 4 depicts the changes in nullclines due to variations of different parameters. Our model revealed that several parameters affect the H3K9me nullclines. Demethylation rate ($\delta_3$, the activity of demethylases like Epe1), methylation spreading rate ($\phi$, chromodomain activity of Clr4 and other chromodomain proteins), affect the H3K9me nullcline (Fig 4A), and basal methylation rate ($\zeta$, the RITS-independent activity of Clr4) affect the desilenced end of H3K9me nullcline (Fig 4B). These parameters are critical for bistability, and this is in agreement with the experimental observation that the loss of Clr4 chromodomain, which is responsible for binding to H3K9me and equivalent to the reduction in $\phi$, leads to variegated bistable heterochromatin establishment [40]. Change in methylation rate ($\varepsilon$, catalytic activity of Clr4) shifts the H3K9me nullcline (Fig 4C), and low $\varepsilon$ can lead to a bistable or monostably desilenced system. On the other hand, transcription rate ($\alpha$, RNAP activity), and siRNA degradation rate ($\delta_2$, Eri1 activity) affect the

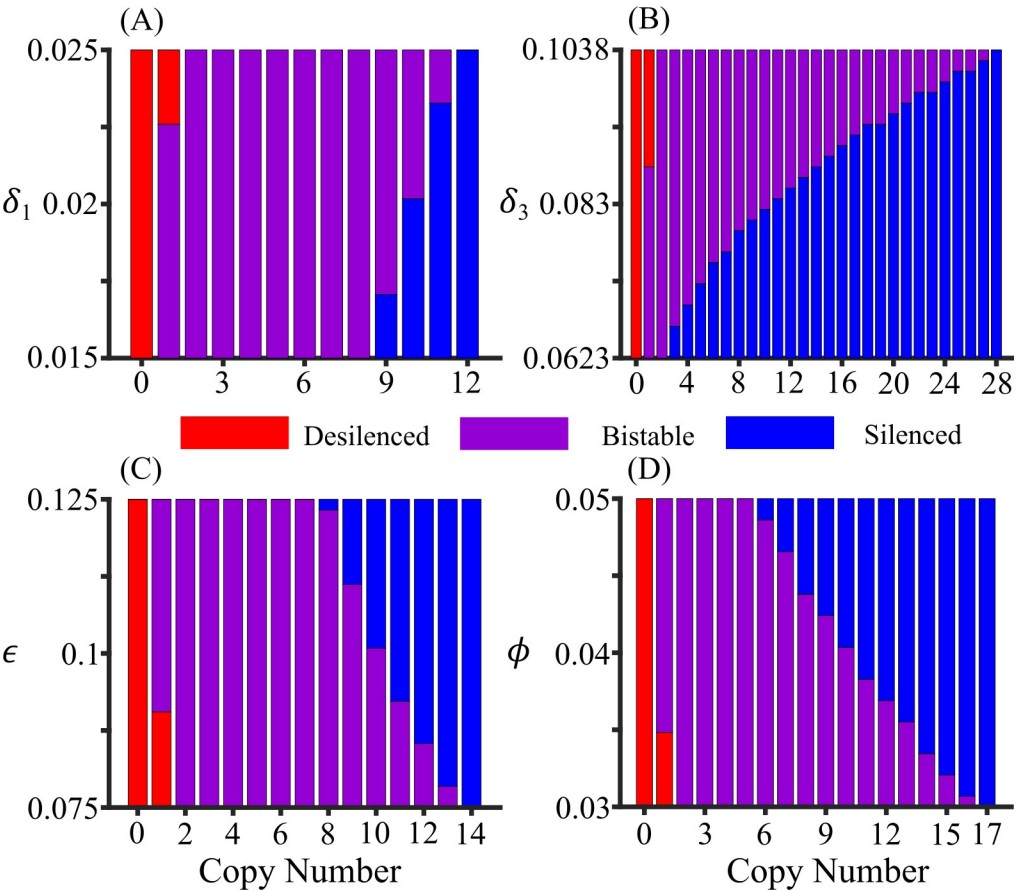

**Fig 3. Parameter sensitivity analyses within 25% variation on default values of a specific parameter.** Colors indicate whether the system is silenced or bistable at the indicated parameter value and copy number. (A) A large RNA degradation rate ($\delta_1$) favors bistability or desilenced state for all copy numbers. All ranges of $\delta_1$ favor bistable state for copy numbers between 2 and 8, favor silenced state for copy number 12 and above, and the range of $\delta_1$ favoring a silenced state increases with an increase in copy number. (B) Like the RNA degradation rate, a higher value of demethylation rate ($\delta_3$) favors bistability or desilenced state. All ranges of $\delta_3$ favor bistable state for copy number 2, favor silenced state for copy number 28 and above, and the range of $\delta_3$ favoring a silenced state increases with the increase in copy number. (C) A small methylation rate ($\varepsilon$) favors bistability or desilenced state. All ranges of $\varepsilon$ favor the bistable state for copy number between 2 and 7, favor the silenced state for copy number 14 and above, and the range of $\varepsilon$ favoring the bistable state decreases with the increase in copy number. (D) Like methylation rate, a lower value of methylation spreading rate ($\phi$) favors bistability or desilenced state. All ranges of $\phi$ favor the bistable state for copy number 2 to 16, favor the silenced state for copy number 17 and above, and the range of $\phi$ favoring the bistable state decreases with copy number.

siRNA nullcline, which primarily affect the desilenced state similar to changing *CN* (Fig 4D). siRNA biogenesis rate ($\gamma$, Dcr1 activity) and RNA degradation rate ($\delta_1$, the activity of ribonucleases like Dhp1) also affect the siRNA nullcline (Fig 4E), and H3K9me can increase with a mutation in exoribonuclease Dhp1 [41]. Hill coefficient ($\rho_2$) and half-maximum methyl concentration ($k_2$) can modulate the slope of the siRNA nullcline, which also controls the number of steady states (Fig 4F–4G), and this parameter determines the cooperativity of H3K9me-dependent transcriptional gene silencing. Hill coefficient ($\rho_1$) and half-maximum methyl concentration ($k_1$) slightly affects the the siRNA nullcline leading to almost no change in the stability of the system (Fig 4H-4I), whereas $\rho_3$ and $k_3$ affect the middle section of the siRNA nullcline leading to bistability from monostable state (Fig 4J-4K). A wide range of Hill coefficients supports bistability (see S5 Fig). These results suggest that the bistability is not a unique

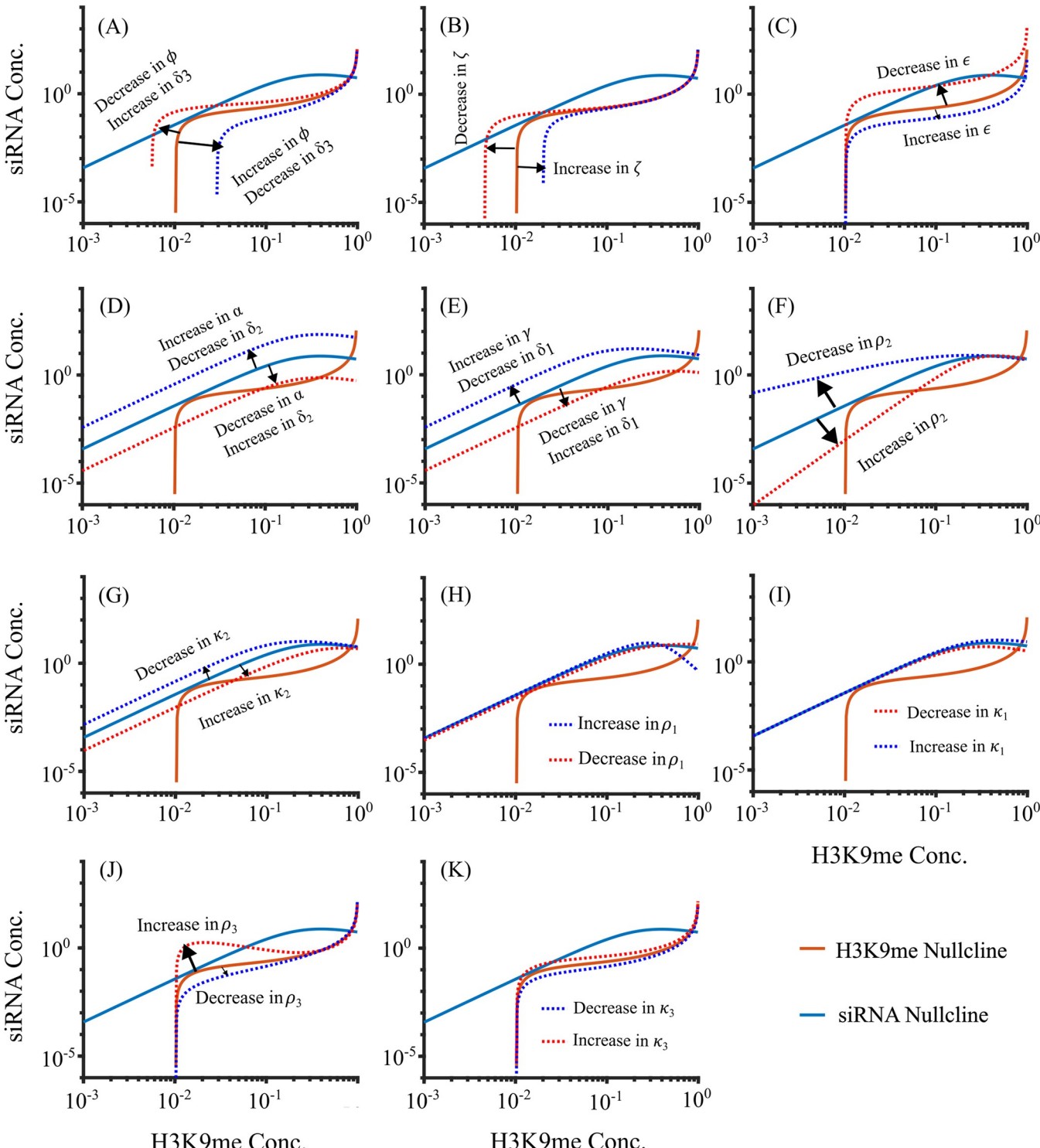

**Fig 4. Changes in the nullclines as a result of changing parameters using QSSA for Copy number 15.** (A) Parameters $\delta_3$ and $\phi$ affect the stability of the desilenced state by modulating the tail and middle section of the H3K9me nullcline. (B) $\zeta$ shifts the tail of the H3K9me nullcline. (C) $\varepsilon$ modulates the middle and head section of the H3K9me nullcline. (D) $\alpha$, and $\delta_2$ can shift the siRNA nullcline similar to changing CN. (E) $\gamma$ and $\delta_1$ shifts the siRNA nullcline. (F) $\rho_2$ changes the slope of the tail of the siRNA nullcline which affects the stability. (G) $k_2$ shifts the tail of the siRNA nullclines. (H) $\rho_1$ bends the head of the siRNA nullcline with negligible effect on stability. (I) $k_1$ slightly bends the head of the siRNA nullcline with no effect on the number of steady states. (J) $\rho_3$ distorts the shape of the middle section of the H3K9me nullcline affecting the steady states. (K) $k_3$ modulates the middle section of the H3K9me nullcline. $\alpha$ = Transcription Rate. $\varepsilon$ = methylation rate. $\phi$ = methylation spreading rate. $\delta_1$, $\delta_2$, and $\delta_3$ = RNA degradation rate, siRNA degradation rate, and demethylation rate

respectively. $\gamma$ = siRNA biogenesis rate. $\zeta$ = Basal methylation rate. $\rho_1$, $\rho_2$, and $\rho_3$ = Hill coefficient for transcription, siRNA biogenesis, and methylation by siRNA respectively. $k_1$, $k_2$, and $k_3$ = Half maximum methylation for transcription, siRNA biogenesis, and methylation by siRNA respectively.

behavior of parameter values, and the system can exhibit 1) bistability between desilenced and silenced states, 2) monostability of the desilenced state, and 3) monostability of the silenced state.

### The model supports experimental results

To test if our model can recapitulate experimental observations, we changed our parameter values to match known genetic backgrounds. Fig 5A–5B shows the bifurcation diagram after setting various RNAi-associated parameters to zero. For example, spreading defect ($\phi = 0$)

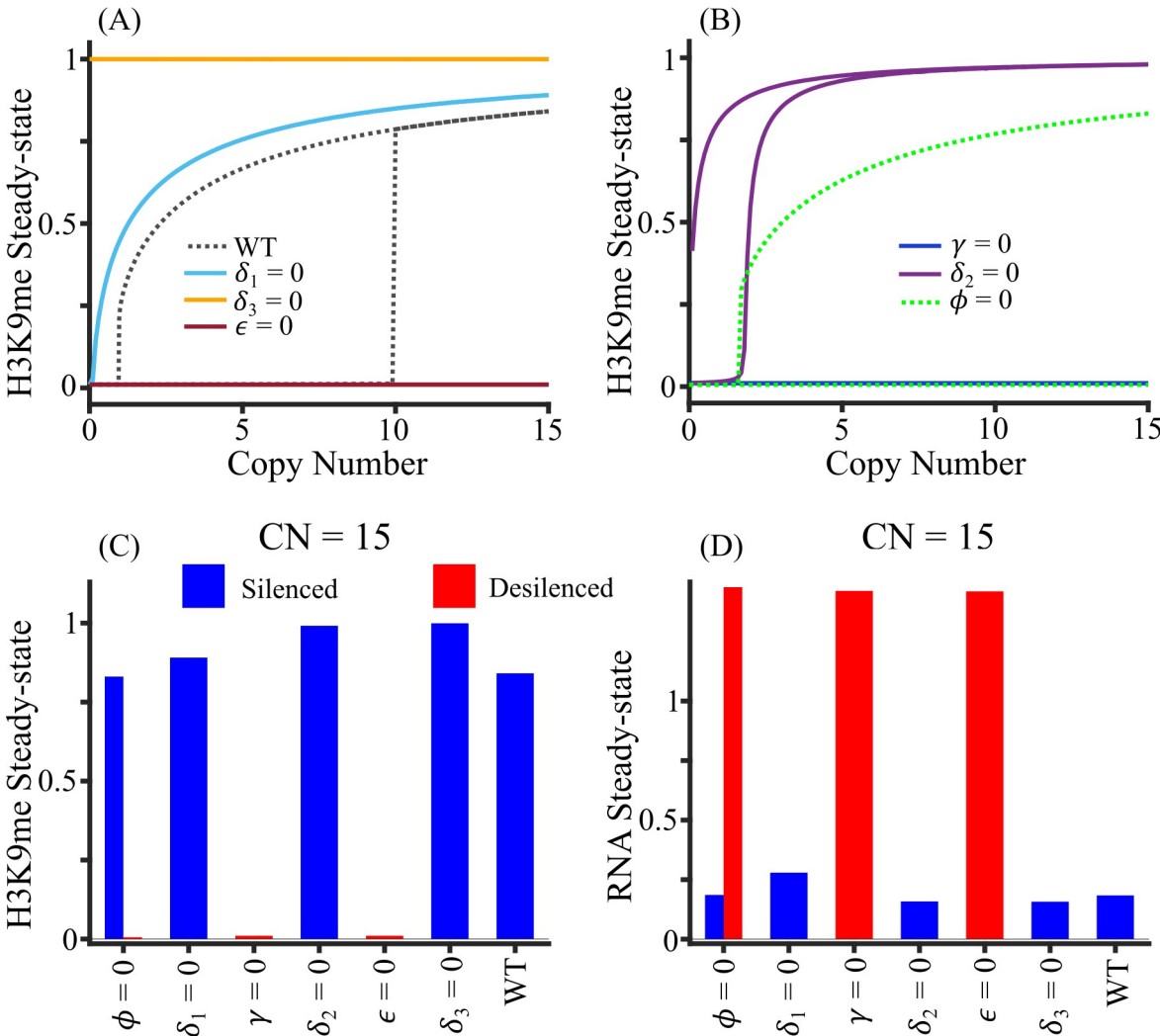

**Fig 5. Comparison of model prediction for deletion experiments.** (A) Bifurcation diagram corresponding to the deletion of the gene encoding RNAi protein, quantified by setting an associated parameter to zero, viz., RNA degradation rate ($\delta_1$), demethylation rate ($\delta_3$ by *epe1*) and methylation rat ($\varepsilon$ by *clr4*). (B) Bifurcation diagrams of siRNA biogenesis ($\gamma$ by *ago1/rdp1/dcr1*), siRNA degradation ($\delta_2$ by *eri1*) and methylation spreading ($\phi$ by *clr4* and other chromodomain proteins) deletions. (C-D) The steady-state concentration of (C) H3K9me and (D) RNA for copy number 15 for deletion of heterochromatin factors. The blue bar represents the silenced state, and the red bar represents the desilenced state.

leads to a bistable state while RNA and siRNA degradation defect favors silencing. Our model predicts that at all copy numbers, the system remains silenced if the demethylation rate is zero and desilenced if the methylation ($\varepsilon$) and siRNA biogenesis ($\gamma$) rates are zero. When the RNA degradation rate ($\delta_1$) is zero, the system is monostable and silenced for a copy number greater than 1. If the siRNA degradation rate ($\delta_2$) is zero, the silenced state is strongly favored for higher copy numbers so that bistability can be observed only for small copy numbers less than 2 (Fig 5B). On the other hand, the system is bistable for copy numbers greater than 2 when the methylation spreading rate is zero (Fig 5B).

Fig 5C–5D shows the steady-state H3K9me and RNA concentrations, respectively, for copy number 15 that would correspond to various gene deletions. Our model recapitulates several experimental results. Spreading defect ($\phi = 0$) corresponds to the loss of Clr4 chromodomain or other chromodomain proteins, which leads to variegated bistable heterochromatin establishment [40]. Consistent with previous reports our model predicts that loss of H3K9me by, for example, *clr4* deletion (equivalent to $\varepsilon = 0$) leads to a loss in centromeric silencing [15,17]. Moreover, our model correctly predicts that loss of siRNA biogenesis pathway, e.g. by deletion of *ago1*, *chp1*, *tas3*, *dcr1*, etc. ($\gamma = 0$), leads to a loss of centromeric silencing [16–18,42]. siRNA degradation defect, $\delta_2 = 0$, is similar to Eri1 deletion, which stabilizes heterochromatin [43]. RNA degradation ($\delta_1$) and demethylation ($\delta_3$) defect can be simulated computationally, but these are processes which are not specific to heterochromatin lncRNAs and H3K9me. For example, Epe1 is the primary known demethylase for H3K9me, but its activity affects not only H3K9me but also H3K36me.

Other studies have reported that the silencing in heterochromatin is variegated, suggesting that the system might be bistable. For example, two distinct outcomes of silenced and desilenced populations were observed with the deletion of histone deacetylase Sir2 and HDAC-associated Clr5 [11,44,45]. In addition, transient overexpression of Swi6 can alter the epigenetic imprint and switch the desilenced state into the silenced state [46]. These observations suggest that our quantitative model can recapitulate several experimental studies on heterochromatin gene silencing and bistability may occur in a wide range of genetic backgrounds.

## Stochastic simulation

ODE solutions represent the average behavior, so we simulated our system stochastically using the Gillespie algorithm [47,48] (see Methods). The maximum number of H3K9me sites was normalized to $nC$, where n is the average number of H3K9me sites per copy and C is the copy number, and RNA and siRNA molecules are rescaled 100-fold to represent individual transcripts. Fig 6 shows the typical dynamics of the system for copy number 15. To minimize the effect of the initial condition, data were sampled from 4000 to 10000 minutes, and the earlier part was excluded as a burn-in period. Cell division can act as a 2-fold dilution and noise from external fluctuations can affect the abundance of RNA, siRNA, and H3K9me, which is proportional to the amount of RNA, siRNA, and H3K9me in the system. Assuming no cell division and no noise in the system, the system remains silenced at CN = 15 (S6 Fig A). In a realistic scenario, we considered two additional variations to our model with external noise and cell division. Fig 6A and S6 Fig E-F show the switching between the silenced and desilenced states where fixed cell division time and noise were incorporated in the simulation.

At each cell division, the number of RNA, siRNA molecules, and H3K9me sites are reduced by half, which S6B Fig depicts. Since H3K9me goes down by 50% at each cell division, faster cell division can destabilize silencing which is shown in S6 Fig. Fraction of silencing at a fixed noise rate (Fig 6B) increases with the increase in cell division time, and S7 Fig A-B show the distribution of H3K9me over time at different cell divisions and fixed noise rate. With the

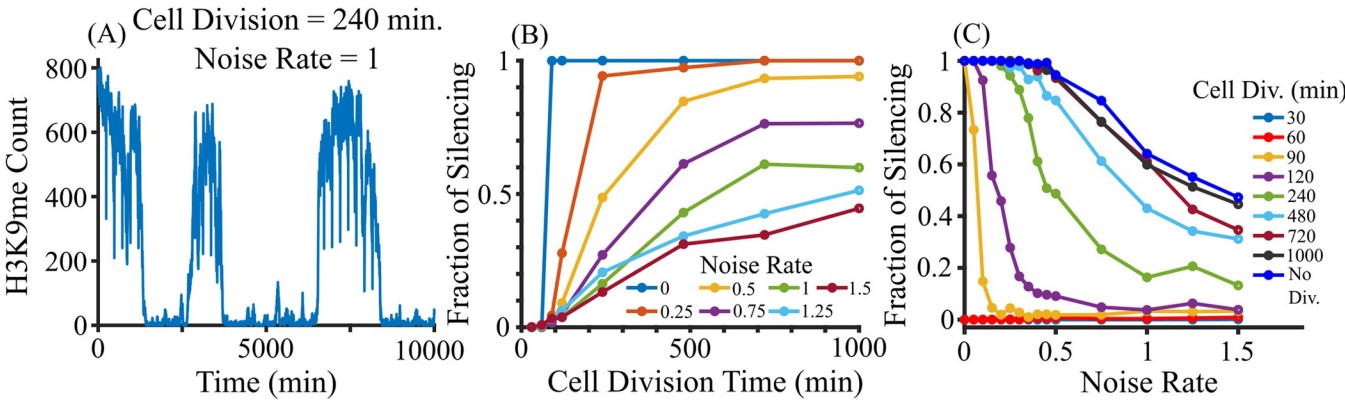

**Fig 6. Stochastic simulation for copy number 15 using Gillespie algorithm.** (A) A stochastic trajectory of a system of cells with an average cell division time of 240 min, and noise rate of 1. (B) Fraction of silencing increases with the increase in cell division for a fixed noise rate. (C) The fraction of silencing decreases with an increase in noise rate for a fixed cell division.

increase in cell division (growth) rate, distribution is favored toward silencing. Noise can facilitate switching from the desilenced state to the silenced state and vice-versa at different cell division rates (S6 Fig), but the fraction of silencing decreases with an increase in noise at fixed cell division (Fig 6C). Our simulation with different growth rates is in agreement with experimental studies. Non- or slow-growing cells exhibit the spreading of H3K9me into euchromatic loci [49,50], suggesting that heterochromatin becomes more stable with a low cell division rate. Our results suggest that the cell division rate plays a critical role in pericentromere silencing, which is a physiologically controllable parameter, and its interplay with cellular noise determines the time evolution of silencing in living cells.

### Alternative model with three-state histone modifications

The heterochromatin assembly not only depends on the machinery for histone methylation but also on histone deacetylases [36,51]. Sir2 and Clr3 are known HDACs associated with heterochromatin assembly. To test the role of HDACs and how HDAC activity shapes the copy-number-dependent methylation within pericentromere, we also developed an alternative model with three histone states: acetylated, unmodified, and methylated. Our alternative five-variable model (with RNA and siRNA) yields results consistent with those of our primary model (see Methods). S8 Fig shows the ODE solutions for this alternative model, demonstrating agreement with our main model's findings. Notably, the existence of two steady states at copy number 5 (S8 Fig A-E) and a single high steady state at copy number 15 (S8 Fig F-J) persists in the alternative model. S9 Fig illustrates the steady-state levels of H3K9me as a function of copy number, and in line with the main model's result, S9 Fig A indicates that WT cells are silenced. With a lower deacetylation rate, desilencing is possible consistent with previous experimental data (S9 Fig B) [32], and methylation is completely lost in the absence of any HDAC activities (S9 Fig C). These indicate that the HDAC activity is another key component of pericentromeric silencing, and the deacetylation rate can also affect the copy-number-dependent pericentromeric repeat silencing.

### Discussion

Here we developed a quantitative model for gene silencing and H3K9me using the fission yeast centromere as a model. This region is one of the best-studied models for heterochromatin formation with many experimental studies testing the function of various associated

chromatin factors. We used the copy number of repeats as an explicit model parameter and showed that the copy number itself can be a critical model parameter that can alter gene silencing. Depending on the copy number, cells can be bistable, with silenced or desilenced pericentromere, which was shown in certain genetic backgrounds experimentally [11,41,45]. Typically, bistability requires nonlinear interactions through various feedback [52] by nonlinear methylation-dependent transcription and siRNA biogenesis in our model. Although we showed how the variation in each parameter affects the steady-state behaviors, the full model in the cell involves many other proteins whose functions are not well known. We assumed that each copy acts independently, but it may be possible that the chromatin state of one repeat preferentially affects the nearest neighbors. In addition, the boundary elements of heterochromatin can affect each repeat in a distant-dependent manner [53,54]. 3D compaction of the genome may further limit enzyme accessibility which facilitates the formation of stable chromatin domains [55], in this case forming a block of silenced repeats,

Our results suggest that RNAi may operate with a threshold copy number for the silencing and target genomic regions with multiple repeats like the pericentromeric repeats, and this leaves the euchromatic loci unaffected by this mechanism. Although few known viruses can infect yeast species, one of the functions of RNAi is a defense against viruses or transposable elements [56,57]. RNAi-mediated copy-number-dependent silencing would be effective against viruses or transposable elements if they generate multiple copies. Therefore, unlike sequence-dependent defense mechanisms like CRISPR [58], RNAi-based silencing might work as a natural sensor and a sequence-independent defense system against foreign genetic elements of multiple copies. The threshold copy number and systemic bistability often require strong cooperativity, which in the case of fission yeast is incorporated by the H3K9me-mediated transcriptional gene silencing. There may exist a threshold copy number with weaker cooperativity like RNAi against multiple copies of dsRNA fragments in the absence of chromatin-based feedback. On the other hand, such a threshold copy number also implies that RNAi-mediated silencing is a mechanism to protect the unique euchromatin part of the genome against silencing. We also demonstrate that HDAC activity is required for this silencing, which may be another tunable feature in addition to RNA-mediated silencing.

Our findings underscore the significance of repeat copy numbers in heterochromatin. While manipulating pericentromeric repeats experimentally poses challenges, a few studies have successfully demonstrated significant alterations in the structure of pericentromeres. Insertion of additional repeat copies with dg-containing minichromosome can lead to the enrichment of H3K9me2 in the exogenous dg copies even in the absence of some heterochromatin factors [26]. A recent study showed that cells with fewer dg/dh copies can maintain H3K9me, but genomic instability increases with the loss of pericentromeric repeat copies, suggesting that heterochromatin function may depend on the copy number [59]. Our models may be applied to other repeat elements in the genome such as telomeres and rDNA clusters as they may undergo copy-number-dependent chromatin-mediated silencing with additional silencing machineries. Specifically, rDNA clusters are up to 50% different in size from isogenic cells [60], and the disruption of the telomere cap complex can lead to rearrangements among telomeres and rDNAs [61]. Such variation in rDNA repeats is due to a high recombination rate, which can be repressed by H3K9me [62,63]. Several studies in other organisms have shown that heterochromatin repeats like rDNA may act as a sensor to monitor genome instability and induce senescence [64,65]. Unlike pericentromere, cells need some copies of active rDNA, and a dynamic interplay between silencing and recombination can alter the rDNA cluster copy number in adaptation to a fluctuating environment.

The copy number variation within heterochromatin is found in many organisms. In humans, the satellite DNA is organized in long arrays in pericentromeric heterochromatin,

and the copy number increases with age [66]. Additionally, human satellite II can massively expand in solid tumors, and this progressive elongation of pericentromeric regions is specific to tumors [4,67]. Duplication of pericentromeric repeats was also observed in the laboratory mouse, suggesting that such expansion is not restricted to primates [68]. Fly genome also shows a considerable variation in satellite DNA abundance [69] where the change in abundance is correlated in various isolated populations. Therefore, pericentromeric repeats may be highly dynamic and functionally important for cellular homeostasis. Our quantitative model shows that such copy number variation among transcriptionally repressed repeats can alter gene silencing and chromatin state. A more complicated quantitative model would be required to assess the gene silencing on pericentromere in higher eukaryotes with additional regulatory machinery for gene silencing like DNA methylation.

## Materials and methods

### Mathematical modeling

Our mathematical model tracks the concentration of lncRNA, siRNA, and H3K9me (Fig 1). The model includes three first-order ODEs.

$$\dot{x}_1 = \frac{C\alpha}{1 + \left(\frac{x_3}{\kappa_1}\right)^{\rho_1}} - \delta_1 x_1 - \frac{\gamma x_1 \left(\frac{x_3}{\kappa_2}\right)^{\rho_2}}{1 + \left(\frac{x_3}{\kappa_2}\right)^{\rho_2}} \tag{1}$$

$$\dot{x}_2 = \frac{\gamma x_1 \left(\frac{x_3}{\kappa_2}\right)^{\rho_2}}{1 + \left(\frac{x_3}{\kappa_2}\right)^{\rho_2}} - \delta_2 x_2 \tag{2}$$

$$\dot{x}_3 = \frac{\epsilon x_2 (1 - x_3) \left(\frac{x_3}{\kappa_3}\right)^{\rho_3}}{1 + \left(\frac{x_3}{\kappa_3}\right)^{\rho_3}} - \delta_3 x_3 + \phi(1 - x_3)x_3 + \zeta(1 - x_3) \tag{3}$$

In Eqs (1–3), $x_1, x_2$, and $x_3$ are the concentrations of RNA transcripts, siRNA, and H3K9me, respectively, and $x_3$ is normalized with a maximum value of 1. $C$ represents the copy number of pericentromeric repeats which is an explicit parameter of the model. Each copy can be transcribed independently, and the overall maximum transcription is linearly proportional to $C$ in the absence of methylation. The transcription rate of RNA, as given by the first term in Eq (1), decreases nonlinearly with the increase in H3K9me concentration as a Hill function, with Hill coefficient $\rho_1$, and half-maximum H3K9me concentration $k_1$. RNA can be degraded at a rate $\delta_1$, or turn into siRNAs in a methylation-dependent manner. siRNA biogenesis rate is represented as a Hill function which is nonlinearly correlated with methylation, with $\rho_2$ as the Hill coefficient and $k_2$ as the half-maximum methylation. Eq (2) describes the change in siRNA concentration which includes the siRNA biogenesis and degradation marked by biogenesis rate $\gamma$ and degradation rate $\delta_2$. The first term in Eq (3) represents the siRNA-mediated methylation, which is proportional to siRNA concentration and available unmethylated H3K9 (1-$x_3$), with the rate that increases non-linearly with H3K9me concentration as a Hill function of the Hill coefficient $\rho_3$ and half-maximum methylation $k_3$. The spreading of methylation is shown by the third term in Eq (3), where $\phi$ is the spreading rate. Methylation other than from the RNAi mechanism is incorporated as basal methylation with rate $\zeta$. $\delta_3$ is the demethylation rate. Here we use $\rho_1 = 1$, $\rho_2 = 2$, and $\rho_3 = 1$, but the bistability occurs for a wide variety of Hill

coefficients. S5 Fig shows the effect of Hill coefficients on bistability, and bistability is possible for different Hill coefficients.

## Solutions of ODE

The system of Eqs (1–3) was solved in MATLAB (R2019b) [70] using an ode45 solver for 24 different initial RNA, siRNA, and H3K9me concentrations. The values of the parameters used for wild-type cells are listed in S2 Table, and 24 initial conditions used in the model are listed in S3 Table.

## Bifurcation diagram

To obtain a bifurcation diagram, Eqs (1–3) were solved for different values of copy numbers ranging from 1 to 20, applying the initial conditions listed in S3 Table for each copy number to 25,000 minutes. All 24 H3K9me steady states for each copy number were plotted to decide how many clusters they group for Fig 2. The system becomes completely silenced for copy numbers 11 and above. Therefore, the k-means algorithm [71], where k = 2, was applied to divide all steady-states into two clusters and get the average of each cluster. If the absolute difference between the average of one cluster and that of the other is smaller than 0.085, they were monostable and bistable if greater than 0.085. Finally, the average steady state of each cluster was plotted against the corresponding copy number. S2 Fig depicts the fraction of silenced and desilenced states obtained after solving Eqs (1–3) for 1000 random initial conditions up to copy number 20. Fractions of both silenced and desilenced states were obtained for copy numbers 1 to 11, where the system is bistable.

## Quasi-steady state approximation (QSSA)

S1 Fig shows that RNA reaches steady-state much faster than siRNA, therefore we can approximate the full system where RNA is in equilibrium with respect to siRNA and H3K9me at any given time. This can be done by setting the left-hand side of Eq (1) to zero and solving for $x_1$, we obtain:

$$x_1 = \frac{C\alpha}{1 + \left(\frac{x_3}{\kappa_1}\right)^{\rho_1}} \frac{1 + \left(\frac{x_3}{\kappa_2}\right)^{\rho_2}}{\delta_1\left(1 + \left(\frac{x_3}{\kappa_2}\right)^{\rho_2}\right) + \gamma\left(\frac{x_3}{\kappa_2}\right)^{\rho_2}} \tag{4}$$

Substituting Eq (4) into Eq (2) leads to:

$$\dot{x}_2 = \frac{\gamma C\alpha \left(\frac{x_3}{\kappa_2}\right)^{\rho_2}}{\left(1 + \left(\frac{x_3}{\kappa_1}\right)^{\rho_1}\right)\left(\delta_1\left(1 + \left(\frac{x_3}{\kappa_2}\right)^{\rho_2}\right) + \gamma\left(\frac{x_3}{\kappa_2}\right)^{\rho_2}\right)} - \delta_2 x_2 \tag{5}$$

Setting LHS of Eq (5) and Eq (3) equal to zero gives the equations of siRNA and H3K9me nullclines respectively.

$$x_2 = \frac{\gamma C\alpha \left(\frac{x_3}{\kappa_2}\right)^{\rho_2}}{\delta_2\left(1 + \left(\frac{x_3}{\kappa_1}\right)^{\rho_1}\right)\left(\delta_1\left(1 + \left(\frac{x_3}{\kappa_2}\right)^{\rho_2}\right) + \gamma\left(\frac{x_3}{\kappa_2}\right)^{\rho_2}\right)} \tag{6}$$

$$x_2 = \frac{\left(1 + \left(\frac{x_3}{\kappa_3}\right)^{\rho_3}\right)(\delta_3 x_3 - \phi(1-x_3)x_3 - \zeta(1-x_3))}{\epsilon(1-x_3)\left(\frac{x_3}{k_3}\right)^{\rho_3}} \tag{7}$$

Solving Eqs (6–7) leads to steady states, both stable and unstable.

## Sensitivity analysis

To test the sensitivity of a parameter (for example $y$) on copy number-dependent gene silencing, it was varied in the range $0.75p \leq y \leq 1.25p$, $p$ being its default value, keeping all other parameters to their respective default values. Then we calculated the H3K9me steady states as a function of $y$ for a fixed copy number. This enabled us to visualize the range of values of a particular parameter for which the system was desilenced, bistable, and silenced, and thus the critical $y$ at which the transition happened. This process was repeated for copy numbers 1 to 40 and all the parameters. The result is depicted in the bar diagram (Fig 3 and S4 Fig). Each bar shows the range of values for which the system is desilenced, bistable, and silenced for a given copy number. The critical $y$ at which the system changes from bistable to silenced or vice-versa is the border between these two ranges.

## Stochastic simulation

To implement the Gillespie algorithm for Eqs (1–3) for stochastic simulations [47,48] at the discrete molecular level, the maximum number of H3K9me sites was normalized to $nC$, where $n$ is the average number of H3K9me sites for methylation per copy, and $C$ is the copy number. Each histone can be independently methylated or demethylated. RNA and siRNA molecules in Eqs (1) and (2) were rescaled 100-fold to represent individual transcripts.

Cell division time was fixed, and cell growth rate was assumed to be linear in time, and simulated as $V = Kt + V_0$, where $K = \frac{V_{max} - V_{min}}{Cell\ division\ time}$ and $V_0$ is the minimum or initial volume after each cell division. Cell volume was increased whenever a reaction occurred, and it was halved when the time approached the next cell division. The maximum and minimum cell volume was taken to be $\frac{4}{3}$ and $\frac{2}{3}$ respectively so that the average cell volume is 1. Consistent with known biology, RNA, siRNA, and H3K9me were divided into two daughter cells at each cell division. The number of RNA, siRNA molecules, and H3K9me sites was increased or decreased by a step size of 1 after each reaction whose occurrence depends on its rate.

The intrinsic noise was considered as a random walk per particle, whose total rate is proportional to either the number of H3K9me sites or RNA and siRNA molecules in the system. e.g., $r_1 = \beta x_1$, $r_2 = \beta x_2$, and $r_3 = \beta x_3$ are three reaction rates for the noise of RNA, siRNA, and H3K9me respectively. $\beta$ is termed as "Noise Rate" in this paper. Number of $x_1$, $x_2$, and $x_3$ are randomly increased or decreased with step size 1 whenever reactions $r_1$, $r_2$, or $r_3$ are selected respectively.

To calculate the silencing fraction, 50 simulations were run to get the H3K9me count over time at a fixed cell division and noise rate. The average silencing fraction of those 50 simulations was taken and plotted at various noise rates and fixed cell division. For the plot of the probability distribution of H3K9me over time, the H3K9me count was taken from all 50 simulations. Even though the simulation was run for 10000 minutes, data up to 4000 minutes were discarded as burn-in.

### Alternative model with two-step processes for methylation

Our main model tracks the level of methylation, but the actual histone states also include acetylation, which is a key step in the heterochromatin assembly. To incorporate this, we have developed an alternative model that considers two additional variables, specifically acetylated H3K9 and unmodified H3K9 [36,72]. In this model, we have assumed that only unmodified H3K9 can be either methylated or acetylated and vice-versa. Also, we assume that new histones are acetylated [35]. In this model, $x_1$, $x_2$, $x_3$, $x_4$, and $x_5$ represent the RNA and siRNA concentrations, the fractions of methylation, acetylated, and unmodified histones, respectively.

$$\dot{x}_1 = \frac{C\alpha}{1 + \left(\frac{x_3}{\kappa_1}\right)^{\rho_1}} - \delta_1 x_1 - \frac{\gamma x_1 \left(\frac{x_3}{\kappa_2}\right)^{\rho_2}}{1 + \left(\frac{x_3}{\kappa_2}\right)^{\rho_2}} \tag{8}$$

$$\dot{x}_2 = \frac{\gamma x_1 \left(\frac{x_3}{\kappa_2}\right)^{\rho_2}}{1 + \left(\frac{x_3}{\kappa_2}\right)^{\rho_2}} - \delta_2 x_2 \tag{9}$$

$$\dot{x}_3 = \frac{\epsilon x_2 x_5 \left(\frac{x_3}{\kappa_3}\right)^{\rho_3}}{1 + \left(\frac{x_3}{\kappa_3}\right)^{\rho_3}} - \delta_3 x_3 + \phi_1 x_3 x_5 + \zeta_1 x_5 - \xi x_3 \tag{10}$$

$$\dot{x}_4 = \zeta_2 x_5 - \delta_4 x_4 + \xi - \xi x_4 \tag{11}$$

$$\dot{x}_5 = -\left(\frac{\epsilon x_2 x_5 \left(\frac{x_3}{\kappa_3}\right)^{\rho_3}}{1 + \left(\frac{x_3}{\kappa_3}\right)^{\rho_3}} - \delta_3 x_3 + \phi_1 x_3 x_5 + \zeta_1 x_5\right) - (\zeta_2 x_5 - \delta_4 x_4) - \xi x_5 \tag{12}$$

Eqs (8) and (9), responsible for tracking the changes in RNA and siRNA concentrations respectively, are consistent with Eqs (1) and (2) mentioned in the three-variables model. Eq (10), which details the variation in H3K9me concentration and is mathematically similar to Eq (3), distinguishes itself from Eq (3) by addressing the methylation acting on unmodified H3K9, denoted by $x_5$. $\xi$ represents the rate of histone turnover, and we assume that newly incorporated histones are acetylated [35]. $\zeta_2$ and $\delta_4$ represent the rate of acetylation and deacetylation, respectively. Only unmodified hisones can be methylated or acetylated.

The parameter referenced and the initial conditions applied for solving ODE Eqs (8–12) can be found in S4 and S5 Table respectively. The starting levels of methylated, acetylated, and unmodified H3K9 were selected to ensure their total equals one. The parameters used in Eqs (8–9) align with those in Eqs (1–2).

We have used Matlab R2019b and the model codes are included as a supporting file (https://figshare.com/articles/media/Untitled_Item/24599379).

### Supporting information

**S1 Fig.** ODE solution of Eqs 1–3 for (A, C, E) CN = 15 and (B, D, F) CN = 5 for the same initial condition. (A, C, E) When CN = 15, the system is silenced, which is represented by high steady-state H3K9me concentration. (B, D, F) when CN = 5, the system is desilenced, which is represented by low steady-state H3K9me concentration.
(TIFF)

**S2 Fig. Fraction of silenced and desilenced states obtained for 1000 different initial conditions.** Bistable region is from CN = 1 to CN = 11.
(TIFF)

**S3 Fig.** Comparison of time evolutions of (A) siRNA, and (B) H3K9me between full model and QSSA at CN = 15. Both siRNA (A) and (B) H3K9me concentrations evolve at similar rates between the full model and QSSA.
(TIFF)

**S4 Fig. Range of parameters (within 25% of their default values) facilitating desilenced, silenced, or bistable states at different copy numbers.** Other than one parameter, all other parameters are held at the default values. High values of (A) $\alpha$, (C) $\gamma$, (D) $\zeta$, (E) $\rho_1$, and (H) $k_1$ favor silencing, whereas high values of (B) $\delta_2$, (F) $\rho_2$, (G) $\rho_3$, (I) $k_2$, and (J) $k_3$ favor bistability or desilencing. $\alpha$ = Transcription Rate, $\delta_2$ = siRNA degradation rate, $\gamma$ = siRNA biogenesis rate, $\zeta$ = Basal methylation rate, $\rho_1$, $\rho_2$, and $\rho_3$ = Hill coefficient for transcription, siRNA biogenesis, methylation by siRNA respectively. $k_1$, $k_2$, and $k_3$ = Half maximum methylation for transcription, siRNA biogenesis, and methylation by siRNA respectively.
(TIFF)

**S5 Fig. Hill coefficient dependence of nullclines.** H3K9me nullcline and siRNA nullcline can intersect at three distinct points if $\rho_1{\geq}1$, $\rho_2{\geq}2$, and $\rho_3{\geq}1$ in the reference parameters.
(TIFF)

**S6 Fig. Stochastic trajectories for copy number 15 for indicated noise level and cell division.** (A) Silenced state is favored for the system with no cell division and noise. (B-C) Cell division can lead to desilencing. (D-F) The H3K9me profile with noise. (G-I), the volume growth with respect to time, for different cell division of the system.
(TIFF)

**S7 Fig. The probability density of H3K9me concentration over time at indicated noise rate and cell division for copy number 15.** Faster cell division favors the desilenced state for a system (A) without noise and (B) with noise. Increase in noise brings the system down to desilenced state (C) without cell division and (D) with cell division.
(TIFF)

**S8 Fig. Five-variable model ODE solutions.** (A-E) ODE solutions when copy number is 5. (F-J) ODE solutions when copy number is 15. Colors indicate the use of different initial conditions.
(PNG)

**S9 Fig. Steady-state H3K9me as a function of copy number for a five-variable model for indicated values for deacetylation rate.**
(PNG)

**S1 Table. Parameters identified by our analysis that facilitate silencing or desilencing.**
(TIF)

**S2 Table. Parameters used for solving differential Eqs (1–3) for WT cell.**
(TIF)

**S3 Table. Initial conditions applied in the simulation.** Methylation concentration is normalized to 1.
(XLSX)

**S4 Table. Parameters used for solving five-variable model differential Eqs (8–12) for WT cell.**
(TIF)

**S5 Table. Initial conditions applied in the simulation of a five-variable model (Eqs 8–12).**
(XLSX)

**S1 Supplementary Code. Code for model and simulation.**
(ZIP)

# Acknowledgments

We are grateful to all the Joh lab members for the critical reading and helpful discussions. We would like to thank Dr. Turbasu Sengupta for his suggestions on figure editing.

# Author Contributions

**Conceptualization:** Mo Motamedi, Richard Joh.

**Data curation:** Puranjan Ghimire, Richard Joh.

**Formal analysis:** Puranjan Ghimire, Richard Joh.

**Funding acquisition:** Richard Joh.

**Investigation:** Puranjan Ghimire.

**Methodology:** Puranjan Ghimire, Mo Motamedi.

**Project administration:** Puranjan Ghimire, Richard Joh.

**Resources:** Richard Joh.

**Supervision:** Richard Joh.

**Validation:** Puranjan Ghimire, Richard Joh.

**Visualization:** Puranjan Ghimire.

**Writing – original draft:** Puranjan Ghimire, Richard Joh.

**Writing – review & editing:** Puranjan Ghimire, Mo Motamedi, Richard Joh.

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
