## [Decision Letter · Decision Letter 0]

4 Nov 2023

Dear Dr. Joh,

Thank you very much for submitting your manuscript "Role of multiple pericentromeric repeats on heterochromatin assembly" for consideration at PLOS Computational Biology.

As with all papers reviewed by the journal, your manuscript was reviewed by members of the editorial board and by several independent reviewers. In light of the reviews (below this email), we would like to invite the resubmission of a significantly-revised version that takes into account the reviewers' comments.

Although the manuscript has been applauded for novelty, there is some criticism about the basic assumptions, which must be addressed before the manuscript can be considered for publication in PLoS Computational Biology.

We cannot make any decision about publication until we have seen the revised manuscript and your response to the reviewers' comments. Your revised manuscript is also likely to be sent to reviewers for further evaluation.

Sincerely,

Jing Chen

Guest Editor

PLOS Computational Biology

Stacey Finley

Section Editor

PLOS Computational Biology

Although the manuscript has been applauded for novelty, there is some criticism about the basic assumptions, which must be addressed before the manuscript can be considered for publication in PLoS Computational Biology.

Reviewer's Responses to Questions

**Comments to the Authors:**

Reviewer #1: The paper is the first modeling paper on heterochromatin formation where siRNA is the prime factor in nucleosome-mediated epigenetics.

The introduction is well written and correctly recapitulates that siRNA is an aspect of obtaining heterochromatin in the pombe mating system in yeast.

The central equations present reasonable simple guesses, although they ignore any spatial constraints of reactions within the studied region.

The discussion is quite weak, without any robust lessons for the reader.

The paper suggests a model for bistable chromatin where the cooperativity comes from methylated H3K9 directing conversions of

RNA to siRNA. Thus it is assumed that double as much methylated nucleosomes increase siRNA production by a factor of 4.

How this should work is unclear. Presumably, two random nucleosomes that a methylated have to act in concert to convert an RNA to an siRNA?

All consequences of this are correctly analyzed, but overall not surprising. Thus any prediction is a direct consequence of the assumed input,

and nothing in the paper demonstrates anything (page 7 " Our study demonstrates that such copy number variation among

transcriptionally repressed repeats can alter gene silencing and genomic stability/instability" is an overstatement)

Fig. 1 is misleading, there is a catalyzing arrow from unmethylated nucleosomes to RNA to siRNA conversion?

The lateral spreading of H3K9me in Fig. 1 is rather poorly represented by the x3*(1-x3) term in eq. 3, as lateral spreading would

change weakly with methylated fraction.

There is no list of used parameters in the main paper, leaving the reader at a loss what a mere 25%

variation in the robustness section was actually accomplished.

Importantly if all hill coefficients were equal to 1, then I predict that there would be no bi-stability.

What is the value of the lifetimes of siRNA and RNA respectively, and is there any justification for this?

The fact that H3K9me represses RNA formation may easily prevent RNA and thus siRNA from being present in heterochromatic state.

This must put constraints on some parameters, associated with the transcriptional activity of RNA and the lifetimes of RNA

(in particular the value of kappa1 must be large relative to x3?)

Overall, it is well-motivated to include siRNA as a contribution to chromatin silencing in s. pombe, but the current study is misleading

in arbitrarily making this particular reaction the main player.

The paper contains too little insights/ideas to justify publication in Plos Computation.

Reviewer #2: This manuscript introduced a mathematical model describing the gene silencing of pericentromeric repeats in which the amount of siRNAs and the level of histone methylation are included as variables, and the copy number of pericentromeric repeats is taken as a parameter. The results show the bistable nature between silenced and desilenced states, and the importance of the copy number on the control of the state. The authors performed the exhaustive bifurcation analysis on the proposed model, and almost no ambiguity is left.

Reviewer #3: In this manuscript the authors describe a mathematical model of histone methylation, focusing on pericentromeric repeats. The authors describe the conditions under which these regions can be stably methylated using a combination of deterministic and stochastic modeling. Overall I found the manuscript well-written, the modeling clearly described, with interesting and thorough analyses, and appropriate conclusions. I have a few minor comments.

I was unclear on whether the pericentromeric repeats are variable across individuals, or within the chromosome itself? The abstract/summary/intro did not specify the type of variability in these repeats. If the variability is within chromosome or across replication events, I'd like to see some discussion by the authors on how this might influence their modeling results, or if the mechanisms of RNAi and siRNA targeting are sensitive to such variability.

I did not see the code provided as a supplemental, I would strongly suggest providing the code, at least for the ODE and Gillespie simulations. I would also like to see the version of Matlab used stated in text in the Materials and Methods section.

The Gillespie simulation showed the impact of cell division time and noise - is this something that is observed or implicated in disease states? If so, I'd like to see this mentioned in the discussion.

I found the title a bit confusing, from what I can gather it seems that methylation influences assembly, but the article doesn't seem to address assembly per se? Perhaps I may be misunderstanding this.

**Have the authors made all data and (if applicable) computational code underlying the findings in their manuscript fully available?**

Reviewer #1: Yes

Reviewer #2: Yes

Reviewer #3: **No: **I did not see the code provided in the supplemental.

PLOS authors have the option to publish the peer review history of their article (what does this mean?). If published, this will include your full peer review and any attached files.

Reviewer #1: No

Reviewer #2: No

Reviewer #3: No
---

## [Decision Letter · Decision Letter 1]

15 Dec 2023

Dear Dr. Joh,

Thank you very much for submitting your manuscript "Role of multiple pericentromeric repeats on heterochromatin assembly" for consideration at PLOS Computational Biology.

As with all papers reviewed by the journal, your manuscript was reviewed by members of the editorial board and by several independent reviewers. In light of the reviews (below this email), we would like to invite the resubmission of a significantly-revised version that takes into account the reviewers' comments.

It is critical for the authors to address the remaining concerns raised by Reviewer 1 about the basic model assumptions.

We cannot make any decision about publication until we have seen the revised manuscript and your response to the reviewers' comments. Your revised manuscript is also likely to be sent to reviewers for further evaluation.

Sincerely,

Jing Chen

Guest Editor

PLOS Computational Biology

Stacey Finley

Section Editor

PLOS Computational Biology

It is critical for the authors to address the remaining concerns raised by Reviewer 1 about the basic model assumptions.

Reviewer's Responses to Questions

**Comments to the Authors:**

Reviewer #1: The revised paper ( PCOMPBIOL-D-23-01233R1) meets the minor comments improvement

but does not alleviate the main concern:

The model is arbitrary and without any other support that siRNA is a player in H3K9 methylation.

The model only considers feedback between histone methylation and siRNA production,

and assumes that this involves some hill coefficients that are larger than 1.

In the answer to previous concerns, the author uses a bunch of references to prove that siRNA

plays a role. This is true, and not disputed by anyone.

However, there is no evidence of any nonlinearity remotely similar to what is used in the present paper.

The authors further claim that their model gives bistability with all hill coefficients equal to 1

and refer to a figure in the supplement that does not support that. Anyway, this should theoretically be impossible.

(as also seen in Fig 4 that considers variations in hill coefficients emphasizing that all lowering of hill coefficients eliminate bistability). Basically, a model without any cooperativity cannot give bistability, and if they can disprove that they would really deserves publication.

It is also of some concern that the whole effect of multiple pleirometric repeats comes in because the number of

RNA produced in eq. 1 enters in equation 2. What is the justification/mechanism for this particular assumption?

This is the only coupling between the repeats, and perhaps the only part of the model

that could be reasonably well argued.

Finally, then why is there exactly zero siRNA produced if there is zero methylated histones, even if there then is plenty of

RNA produced (no leaking term) ?

Overall, although the paper analyzes the equations reasonably well, there is no reason why these quite arbitrary

equations give new insights into the proposed problem. I do not recommend publication.

Reviewer #2: No question unanswered.

Reviewer #3: The authors addressed all of my concerns, I find the manuscript improved.

**Have the authors made all data and (if applicable) computational code underlying the findings in their manuscript fully available?**

Reviewer #1: Yes

Reviewer #2: Yes

Reviewer #3: Yes

PLOS authors have the option to publish the peer review history of their article (what does this mean?). If published, this will include your full peer review and any attached files.

Reviewer #1: No

Reviewer #2: No

Reviewer #3: No
---

## [Decision Letter · Decision Letter 2]

29 Feb 2024

Dear Dr. Joh,

Thank you very much for submitting your manuscript "Role of multiple pericentromeric repeats on heterochromatin assembly" for consideration at PLOS Computational Biology. As with all papers reviewed by the journal, your manuscript was reviewed by members of the editorial board and by several independent reviewers. The reviewers appreciated the attention to an important topic. Based on the reviews, we are likely to accept this manuscript for publication, providing that you modify the manuscript according to the review recommendations.

Reviewer 1 insists on their criticism. I agree with Reviewer 2's opinion that models can have this type of assumptions. However, please try to discuss about the potential pitfalls and future model revisions in light of the criticism raised by Reviewer 1. This will enhance the manuscript.

Sincerely,

Jing Chen

Guest Editor

PLOS Computational Biology

Stacey Finley

Section Editor

PLOS Computational Biology

Reviewer 1 insists on their criticism. I agree with Reviewer 2's opinion that models can have this type of assumptions. However, please try to discuss about the potential pitfalls and future model revisions in light of the criticism raised by Reviewer 1. This will enhance the manuscript.

Reviewer's Responses to Questions

**Comments to the Authors:**

Reviewer #1: The revised paper ( PCOMPBIOL-D-23-01233R2) does little to alliviate my main concern,

that the model is quite arbitrary, and only justified by papers that prove

the importance of methylation via siRNA. However also deacetylation

H3K9 is crucial, acting through the deacetylases Clr3 or Sir2 (see below list of references).

This two-step process has already been included in standard models in the epigenetics field

for the last 15 years (some references below).

I am a big supporter of simple models, so all in favour of removing details. But in this case I

believe that you have thrown the baby out with the water.

Of smaller note, then your argument that eq. 3 gives bistability with constant x2 does not sound true:

The spreading term x3*(1-x3) cannot give bistability, as it never provides a steeper than proportional increase of x3 with x3.

However, there might be some room in parameter space if siRNA dynamics are so fast that x2 is proportional to x3 when inserted in the first term of eq. 3.

In any case, the whole analysis in the paper uses the Hill coefficient rho2=2 without any justification

(How could one possibly imagine a mechanism where double the fraction of methylated nucleosomes

could give four times the number of siRNA. Also, no experimental references were given to support that basic assumption).

Therefore the paper is too misleading to be recommended for publication.

Review of some of the models (some of which specialize in H3K9 in S.Pombe):

O’Kane, Callum J., and Edel M. Hyland. "Yeast epigenetics: the inheritance of histone modification states." Bioscience Reports 39.5 (2019): BSR20182006.

for more details see also:

Dodd, Ian B., et al. "Theoretical analysis of epigenetic cell memory by nucleosome modification." Cell 129.4 (2007): 813-822.

Angel, Andrew, et al. "A Polycomb-based switch underlying quantitative epigenetic memory." Nature 476.7358 (2011): 105-108.

M. Ancona, et al., Competition between local erasure and long-range spreading of a single biochemical mark leads to epigenetic bistability, Phys. Rev. E 101, 042408 (2020).

A. Z. Abdulla et al. Painters in chromatin: a unified quantitative framework to systematically characterize epigenome regulation and memory, Nucleic Acids Res. 50, 9083 (2022).

References showing the requirement for the histone deacetylase Clr3 for heterochromatic gene silencing :

Histone deacetylase homologs regulate epigenetic inheritance of transcriptional silencing and chromosome segregation in fission yeast. Grewal SI, Bonaduce MJ, Klar AJ.Genetics. 1998 Oct;150(2):563-76. doi: 10.1093/genetics/150.2.563.PMID: 9755190

The nucleation and maintenance of heterochromatin by a histone deacetylase in fission yeast. Yamada T, Fischle W, Sugiyama T, Allis CD, Grewal SI.Mol Cell. 2005 Oct 28;20(2):173-85. doi: 10.1016/j.molcel.2005.10.002.PMID: 16246721

Three additional linkage groups repress transcription and meiotic recombination in the mating-type region of Schizosaccharomyces pombe.

Thon G, Cohen A, Klar AJ.Genetics. 1994 Sep;138(1):29-38. doi: 10.1093/genetics/138.1.29.PMID: 8001791

Mutations in rik1, clr2, clr3, and clr4 genes asymmetrically derepress the silent mating-type loci in fission yeast. Ekwall K, Ruusala T.Genetics. 1994 Jan;136(1):53-64. doi: 10.1093/genetics/136.1.53.PMID: 8138176)

References showing a requirement for the histone deacetylase Sir2 in heterochromatin:

Sir2 regulates histone H3 lysine 9 methylation and heterochromatin assembly in fission yeast. Shankaranarayana GD, Motamedi MR, Moazed D, Grewal SI.Curr Biol. 2003 Jul 15;13(14):1240-6. doi: 10.1016/s0960-9822(03)00489-5.PMID: 12867036

Distinct roles for Sir2 and RNAi in centromeric heterochromatin nucleation, spreading, and maintenance. Buscaino A, Lejeune E, Audergon P, Hamilton G, Pidoux A, Allshire RC.EMBO J. 2013 May 2;32(9):1250-64. doi: 10.1038/emboj.2013.72. Epub 2013 Apr 9.PMID: 23572080

Sir2 is required for Clr4 to initiate centromeric heterochromatin assembly in fission yeast. Alper BJ, Job G, Yadav RK, Shanker S, Lowe BR, Partridge JF.EMBO J. 2013 Aug 28;32(17):2321-35. doi: 10.1038/emboj.2013.143. Epub 2013 Jun 14.PMID: 23771057

(Comment in: Histone deacetylases govern heterochromatin in every phase. Murakami Y.EMBO J. 2013 Aug 28;32(17):2301-3. doi: 10.1038/emboj.2013.154. Epub 2013 Jul 5.PMID: 23832177)

Reviewer #4: In this manuscript, Ghimire et al. present a mathematical model for the relationship between pericentromeric repeat copy number and heterochromatin assembly. Their work introduces an important new variable into a model for silencing in pericentromeric DNA repeats. Previous modeling attempts have relied on histone modification feedback loops, the formation of condensates, and competition between activating and repressing histone modifications. Taking copy number of repeats into account is a valuable new addition that should be of general interest to the field.

I have also gone over the comments of reviewer #1 which raises some important points about the modeling effort. I feel like the authors have provided reasonable answers to the reviewer's concerns. While the model contains some arbitrary parameters, it also makes predictions regarding the importance of copy number that can be experimentally tested in the future in S. pombe and other systems. As such, it is likely to stimulate new thinking that will impact future modeling efforts dealing with epigenetic inheritance and bistability as well as new lines of experimentation.

The mathematical modeling also relies on a deep knowledge of RNAi-mediated heterochromatin formation and is consistent with the well-established genetic requirements for this process in S. pombe. As noted above, the predictions of the model regarding copy number and its relationship to H3K9me and siRNA concentration are testable (although beyond the scope of the present study) and will stimulate future studies.

The authors should modify their title to indicate their study is a modeling effort. For example, “A mathematical model for the role of …. “

I support publication.

**Have the authors made all data and (if applicable) computational code underlying the findings in their manuscript fully available?**

Reviewer #1: **No: **I did not find the analysis supporting their claim that bistability could be obtained with all hill coefficients equal to 1.

Reviewer #4: Yes

PLOS authors have the option to publish the peer review history of their article (what does this mean?). If published, this will include your full peer review and any attached files.

Reviewer #1: No

Reviewer #4: No

Figure Files:

Data Requirements:

Reproducibility:

References:

---

## [Editor Report · Decision Letter 3]

27 Mar 2024

Dear Dr. Joh,

We are pleased to inform you that your manuscript 'Mathematical model for the role of multiple pericentromeric repeats on heterochromatin assembly' has been provisionally accepted for publication in PLOS Computational Biology.

Best regards,

Jing Chen

Guest Editor

PLOS Computational Biology

Stacey Finley

Section Editor

PLOS Computational Biology

---

## [Editor Report · Acceptance letter]

3 Apr 2024

PCOMPBIOL-D-23-01233R3 

Mathematical model for the role of multiple pericentromeric repeats on heterochromatin assembly

Dear Dr Joh,

I am pleased to inform you that your manuscript has been formally accepted for publication in PLOS Computational Biology. Your manuscript is now with our production department and you will be notified of the publication date in due course.

With kind regards,

Zsofia Freund
